# Combined Treatment of Sodium Butyrate and Bromelain Enhanced Anticancer Effects in Colorectal Cancer Cell Lines: A Promising Therapeutic Approach

**DOI:** 10.3390/ijms26199803

**Published:** 2025-10-08

**Authors:** Rocío Olivera-Salazar, Pedro Villarejo Campos, Rocío Barrueco Gutiérrez, Luz Vega-Clemente, Luis Javier Serrano, Soledad García Gómez-Heras, Damián García-Olmo, Mariano García-Arranz

**Affiliations:** 1New Therapies Laboratory, Health Research Institute-Fundación Jiménez Díaz University Hospital (IIS-FJD), Avda. Reyes Católicos, 2, 28040 Madrid, Spain; rocio.olivera@iis-fjd.es (R.O.-S.); rocio.barrueco@iis-fjd.es (R.B.G.); luz.vega@iis-fjd.es (L.V.-C.); luis.serrano@quironsalud.es (L.J.S.); damian.garcia@quironsalud.es (D.G.-O.); 2Department of Surgery, Fundación Jiménez Díaz University Hospital (FJD), 28040 Madrid, Spain; pedro.villarejo@quironsalud.es; 3Department of Basic Health Science, Faculty of Health Sciences, Rey Juan Carlos University, 28922 Alcorcón, Spain; soledad.garcia@urjc.es; 4Department of Surgery, Universidad Autónoma de Madrid, 28034 Madrid, Spain

**Keywords:** colorectal cancer, sodium butyrate, bromelain, combined treatment, synergistic antitumor effect

## Abstract

Colorectal cancer (CRC) is one of the most prevalent and lethal cancers worldwide, with few effective treatment options and substantial associated side effects. As a result, there is growing interest in therapeutic alternatives that reduce toxicity. Natural compounds such as sodium butyrate (NaB), a microbial metabolite of dietary fiber, and bromelain, a proteolytic enzyme from pineapple, have shown individual anticancer properties. However, their combined effect in CRC remains underexplored. This study investigates the synergistic potential of NaB and bromelain in colorectal cancer cell lines, focusing on their ability to inhibit proliferation, induce apoptosis, and modulate key molecular pathways. Findings reveal that co-treatment enhances antitumor activity *in vitro*, suggesting a promising and safer therapeutic strategy for CRC.

## 1. Introduction

Colorectal cancer (CRC) remains one of the most prevalent and lethal malignancies worldwide, representing a major public health challenge [1,2,3]. Despite advancements in surgical techniques and chemotherapeutic strategies, the prognosis for advanced CRC remains poor, with limited therapeutic options and significant treatment-related toxicity [4].

Given these limitations, there is growing interest in natural or diet-derived compounds as safer anticancer alternatives. Acquired drug resistance in CRC remains a major challenge, driving the need for more effective therapeutic strategies [5].

Among the available non-toxic options, dietary fiber is known to protect against CRC, largely due to its fermentation by gut bacteria into short-chain fatty acids (SCFAs) such as sodium butyrate (NaB). NaB stands out for its dual role: serving as a primary energy source for colon cells and acting as a potent histone deacetylase (HDAC) inhibitor (HDACi), with strong tumor-suppressive effects [6,7,8,9,10,11,12]. However, levels of certain NaB-producing bacteria most notably *Clostridium butyricum*, a known probiotic, are often found to be reduced in CRC patients.This microbial imbalance may lead to reduced colonic NaB concentrations, potentially compromising its protective and regulatory effects. Since NaB acts as HDACi, reduced availability may impair epigenetic regulation of key genes involved in cell cycle control and apoptosis [13].

Given the reversible nature of epigenetic modifications, HDACs have emerged as key therapeutic targets in cancer. HDAC inhibition by compounds like NaB promotes beneficial gene expression changes that support apoptosis, induce autophagy, cell cycle arrest, inhibited cancer cell growth, and reduced metastasis, highlighting its potential for CRC prevention and treatment [6,14,15,16,17,18,19].

Although several HDAC inhibitors are currently under clinical investigation, either as monotherapies or in combination with other agents, their clinical application remains limited due to modest efficacy when used alone. This has led to increased interest in exploring novel combinatory therapies that maximize anticancer potential while maintaining safety [6].

In this context, bromelain, a cysteine protease complex derived primarily from pineapple (*Ananas comosus*), has gained attention for its diverse biological activities. Bromelain exhibits anti-inflammatory, immunomodulatory, and anticancer properties, with demonstrated abilities to suppress tumor cell proliferation, induce apoptosis, and enhance the bioavailability of co-administered drugs [20,21,22]. It also modulates inflammation by degrading inflammatory cytokines and fibrin deposits, inhibiting COX-2 activity, reducing prostaglandin production, and regulating cytokines such as TNF-α and IL-1β, positioning it as a potential alternative or adjunct to conventional anti-inflammatory drugs [23].

Bromelain has been studied in combination with compounds such as N-acetylcysteine, cisplatin, peroxidase, ethanol extract of *Olea europaea* leaves (EOLE), and curcumin in the treatment of various diseases, including cancer. Clinical trials have demonstrated its safety in both animals and humans, notably in formulations such as BromAc^®^ (bromelain plus N-acetylcysteine) [22]. This makes bromelain a promising candidate for evaluating its combination with NaB.

To date, limited research has explored the potential synergistic effects of bromelain and NaB in CRC. This study aims to evaluate their combined anticancer activity on CRC cell lines (HCT116, SW480 and LS174T), focusing on proliferation inhibition, apoptosis induction, and the regulation of gene expression associated with tumor progression, using the normal human colon fibroblast cell line (CCD18-Co) as a control. *In vitro* assays were conducted to assess the efficacy and safety of this combination. Our findings indicate that co-treatment with NaB and bromelain exerts a synergistic antitumor effect *in vitro*, suggesting a promising direction for the development of novel combinatorial therapies for CRC.

## 2. Results

All selected cell lines underwent a preliminary characterization to ensure the reliability of the results. As an essential quality control step, all cell lines were tested for mycoplasma contamination, which is known to alter cell cycle dynamics and compromise experimental accuracy. Following this, the morphology and proliferation profiles of each cell line were evaluated under standard culture conditions to establish baseline characteristics.

### 2.1. Analysis of Cell Morphology

The morphological characteristics of CCD18-Co, HCT116, LS174T, and SW480 cell lines were assessed using bright-field microscopy (Figure 1A–D) and fluorescence staining with DAPI and phalloidin (Figure 1E–H). CCD18-Co cells displayed an elongated, fibroblast-like morphology (Figure 1A) with cells aligned in parallel and a well-organized actin cytoskeleton visible under fluorescence (Figure 1E). HCT116 (Figure 1B,F) and SW480 (Figure 1D,H) cells exhibited mixed morphologies, including both elongated and rounded shapes. In contrast, LS174T cells (Figure 1C,G) displayed a rounded, clustered morphology. All cell cultures were adherent and displayed morphological characteristics consistent with those reported by ATCC, from which they were obtained.

### 2.2. Analysis of Cell Proliferation

On the other hand, cell viability was analyzed using the Deep Blue reagent. All cell lines proliferated during the 96 h of the study (Figure 2). HCT116 and SW480 showed greater proliferation than LS174T, with HCT116 exhibiting the highest proliferation rate. LS174T was the least proliferative cancer cell line, and even showed lower proliferation than the normal colon fibroblast line CCD18-Co.

### 2.3. Evaluation of Synergy Between Sodium Butyrate and Bromelain in Colorectal Cancer Cells

Dose–response curves were generated for sodium butyrate (NaB) and bromelain individually in each cell line. The inhibitory concentration (IC) that reduces cell viability by 25% (IC_25_) was calculated and subsequently used in combination experiments to evaluate potential synergistic effects, since preliminary tests using IC_50_ concentrations resulted in complete cell death.

#### 2.3.1. Dose–Response Curves and IC_25_ Calculation

Cells were treated with a range of concentrations of NaB (1.25–20 mM) and bromelain (10–50 µg/mL) based on values reported in the literature [17,18,19,20]. Cell viability was evaluated using the Deep Blue reagent after 72 h of treatment. The inhibitory concentration required to reduce cell viability by 25% (IC_25_) for each compound was calculated, taking the control cells as 100% cell viability (Figure 3). In the tumor cell lines, both compounds exhibited a sigmoidal dose–response curve. In the normal cell line, while NaB appeared to follow a sigmoidal curve, bromelain showed a U-shaped response. The results showed that the IC_25_ value of NaB for CCD18-Co (17.31 mM) was higher than those for the colorectal cancer cell lines: 1.05 mM for HCT116, 7.33 mM for LS174T, and 3.01 mM for SW480. Conversely, the IC_25_ values for bromelain were relatively similar across cell lines: 19.2 µg/mL for CCD18-Co, 25.01 µg/mL for HCT116, 19.32 µg/mL for LS174T, and 21.43 µg/mL for SW480.

#### 2.3.2. Combined Treatment with IC_25_ Sodium Butyrate and IC_25_ Bromelain

To evaluate potential synergistic effects, the IC_25_ concentrations of NaB and bromelain were combined and applied to colorectal cancer cell lines. This approach allowed assessment of treatment interactions at sublethal doses to avoid complete cell death observed with higher concentrations that allow analyze morphology changes (Figure 4), apoptosis and necrosis (Figure 5), cell cycle (Figure 6) and changes in the gene expression (Figure 7).

Morphological analysis after combination treatment

After treatment, a reduction in the number of attached cells was observed with NaB. In the case of bromelain, the cells detached from the plate and formed clusters, possibly as a strategy to evade the treatment. The combination of both compounds further enhanced these effects (Figure 4).

**Figure 4 ijms-26-09803-f004:**
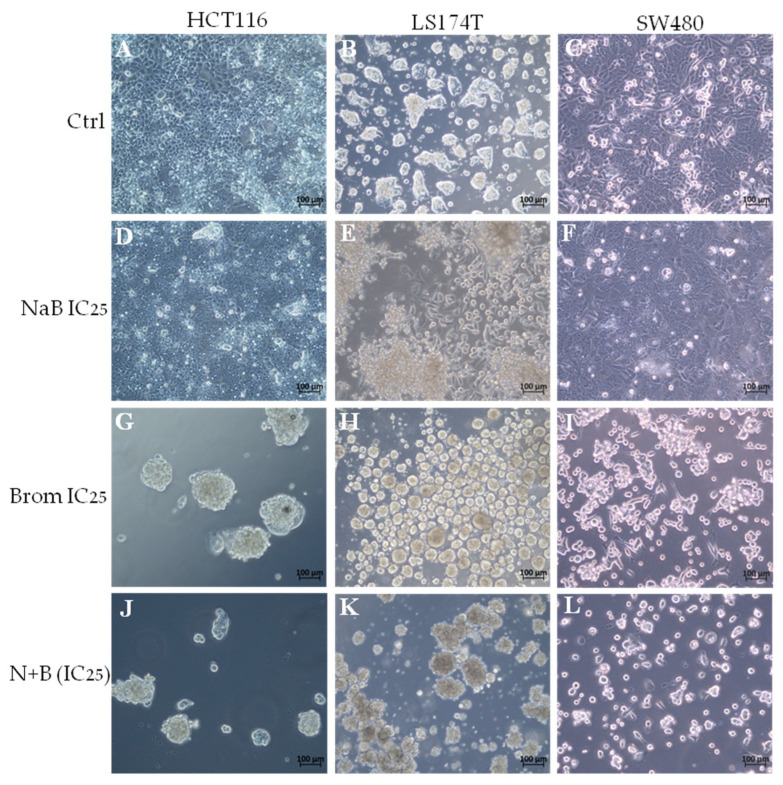
Morphological changes observed under bright-field optical microscopy (20× objective) in colorectal cancer cell lines HCT116 (**A**,**D**,**G**,**J**), LS174T (**B**,**E**,**H**,**K**), and SW480 (**C**,**F**,**I**,**L**) after treatment with the 25% inhibitory concentration (IC_25_) of Sodium Butyrate (NaB IC_25_; **D**–**F**) and Bromelain (Brom IC_25_; **G**–**I**), applied individually or in combination (N + B IC_25_; **J**–**L**), compared to untreated control cells (Ctrl; **A**–**C**) cultured in conventional DMEM medium.

Analysis of Necrosis/Apoptosis Changes Following Combination Treatment

NaB treatment reduced cell viability and increased both early and late apoptosis in HCT116, as well as late apoptosis in SW480 cells, with these effects being significantly enhanced under the combination treatment compared to the single agents (Figure 5). Among the cell lines tested, HCT116 appeared to be the most sensitive (Figure 5B). In LS174T cells, these effects were more moderate, with an increase in early apoptosis observed only with the combined treatment (Figure 5C).

**Figure 5 ijms-26-09803-f005:**
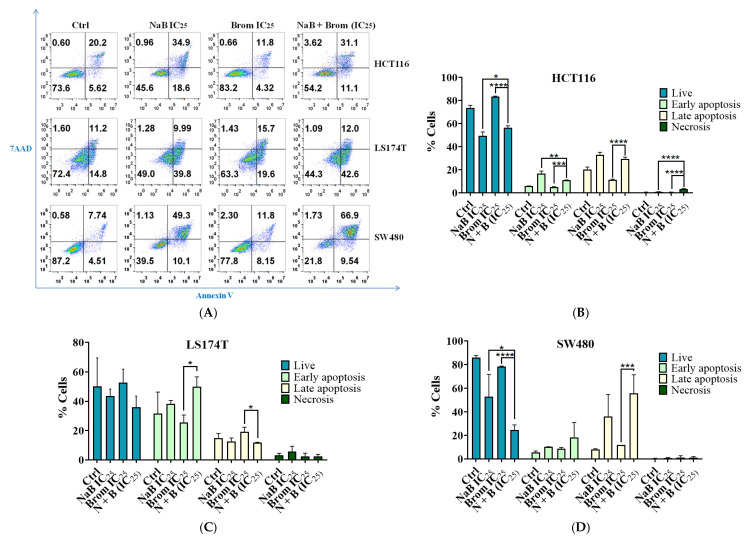
Analysis of apoptosis and necrosis by flow cytometry in colorectal cancer cell lines (HCT116, LS174T, and SW480) after 72 h treatment with IC_25_ concentrations of sodium butyrate (NaB), bromelain (Brom), and their combination. (**A**) Representative flow cytometry plots. Annexin V and 7-AAD staining were used to distinguish viable cells (double negative), necrotic cells (7-AAD positive only), early apoptotic cells (Annexin V positive only), and late apoptotic cells (double positive). (**B**–**D**) Statistical analysis of apoptosis and necrosis in HCT116, LS174T, and SW480 cells, respectively. The statistical analysis highlights the most relevant significant differences observed with the combination treatment. Data are presented as mean ± SD of the proportion of cells from at least three independent experiments. Statistical analysis was performed using one-way ANOVA followed by Tukey’s Post Hoc test. Significance: * *p* < 0.05; ** *p* < 0.01; *** *p* < 0.001; **** *p* < 0.0001; α = 0.05.

Analysis of Cell Cycle Changes Following Combination Treatment

Cell cycle changes were analyzed following the combination treatment in these cell lines (Figure 6A). HCT116 was the most sensitive cell line, showing significant differences in the G0, G1, S, and G2/M phases (Figure 6B). LS174T was the least sensitive, with significant differences detected only in the G1 phase (Figure 6C). Finally, in SW480 cells treated with NaB, there was a significant reduction in the number of cells in the G1 and S phases, which was further enhanced in the combination treatment (Figure 6D).

**Figure 6 ijms-26-09803-f006:**
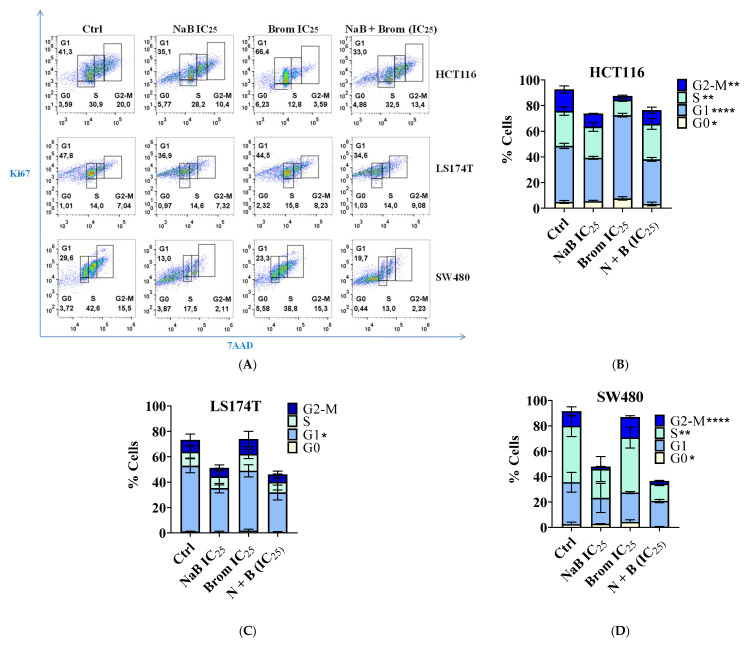
Analysis of cell cycle by flow cytometry in colorectal cancer cell lines (HCT116, LS174T, and SW480) after 72 h treatment with IC_25_ concentrations of sodium butyrate (NaB), bromelain (Brom), and their combination.Ki67 staining was used to assess proliferation, and 7-AAD staining was proportional to DNA content, allowing discrimination of cells in different phases: G0 (Ki67 negative), G1 (Ki67 positive, low 7-AAD), S (Ki67 positive, moderate 7-AAD), G2/M (Ki67 positive, high 7-AAD. (**A**) Representative flow cytometry plots. (**B**–**D**) Statistical analysis of cell cycle phases in HCT116, LS174T, and SW480 cells, respectively. The statistical analysis highlights the most relevant significant differences observed with the combination treatment. Data are presented as mean ± SD of the proportion of cells from at least three independent experiments.Statistical analysis was performed using one-way ANOVA followed by Tukey’s Post Hoc test. Significant differences are indicated with asterisks (* *p* < 0.05, ** *p* < 0.01, *** *p* < 0.001, **** *p* < 0.0001; α = 0.05), representing comparisons between the combination treatment and the individual monotherapy treatments.

Gene expression changes after treatment

After 72 h of IC_25_ combination treatment, colorectal cancer cell lines were harvested to analyze gene expression by RT-PCR. As shown in Figure 7, *CDKN2A* (p16)was upregulated in all cell lines treated with sodium butyrate (NaB), and this effect was further enhanced when combined with bromelain. The increase was statistically significant in SW480 cells, indicating a potential synergistic effect of the combination on cell cycle regulation. For further details, the primer sequences can be found in Table 1 of Section 4.

**Figure 7 ijms-26-09803-f007:**
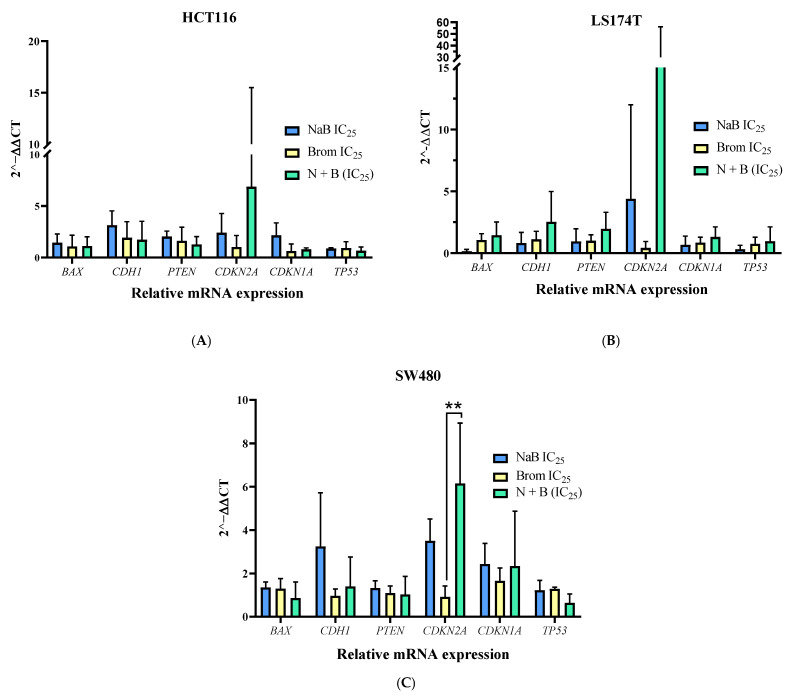
Relative mRNA expression of *BAX*, *CDH1*, *PTEN*, *CDKN2A*, *CDKN1A*, and *TP53* in colorectal cancer cell lines (**A**) HCT116, (**B**) LS174T, and (**C**) SW480 was quantified using RT-qPCR and analyzed using the 2^−ΔΔCt^ method. Expression levels were normalized to *GAPDH* as a housekeeping gene using the ΔCt method, and to the control group (cells cultured in conventional medium) using the ΔΔCt method, after 72 h treatment with sodium butyrate (NaB, IC25), bromelain (Brom, IC25), or their combination. Data are presented as mean ± SD from at least three independent experiments. Statistical analysis was performed using one-way ANOVA followed by Tukey’s Post Hoc test. Significance: * *p* < 0.05; ** *p* < 0.01; *** *p* < 0.001; **** *p* < 0.0001; α = 0.05.

#### 2.3.3. Evaluation of Synergic Doses

The synergy between NaB and bromelain on the inhibition of cell viability was analyzed in HCT116, LS174T and SW480 using the Highest Single Agent (HSA) model, which compares the combined effect of both agents to the effect of the most effective single agent alone. The results (Figure 8) demonstrated synergy (HAS δ-score greater than 0) in all three lines, with LS174T exhibiting the highest degree of synergy (HAS δ-score = 13.004) and SW480 showing the lowest (HAS δ-score = 0.318).

### 2.4. Therapeutic Window-Based Optimal Dose Determination

To identify the optimal treatment concentration, a dose–response matrix was generated for each colorectal cancer cell line (HCT116, SW480, LS174T) and the normal colon cell line (CCD18-Co) using combinations of NaB at concentrations ranging from 1.25 to 20 mM and bromelain at concentrations ranging from 10 to 50 µg/mL. The most toxic dose for the colorectal cancer cells (more than 80% of inhibition), which also demonstrated acceptable biosafety in normal cells (around 25% of inhibition or 75% viability), was then selected to determine the optimal concentration that was combination of 10 mM NaB and 10 µg/mL of bromelain (0.303 mM). With this dose the CCD18-Co has a 27.35% of inhibition while HCT116 has 96.98%, SW480 showed 94.14% and LS174T cell line showed an 84.43% of inhibition (Figure 9).

### 2.5. Biosafety Assessment in Normal Cells

Following the identification of the optimal dose, the cytotoxic effect on normal colon cells (CCD18-Co) was further evaluated (Figure 10). At this concentration, cell viability remained at ~75%, with significant differences observed in early apoptosis compared to untreated controls (Figure 10A,B). Cell cycle analysis revealed a reduction in the proportion of cells in the S phase in the NaB and combination groups compared with bromelain alone and the control group (Figure 10A,C). Additionally, combination treatment induced a marked increase in *CDH1* gene expression (Figure 10D).

### 2.6. Effect of Sequential Treatment on Cell Viability in Normal and Tumor Cell Lines

Initial treatment with the optimal dose resulted in a viability reduction of ~25% in normal human colonic fibroblast cell line(CCD18-Co), ~85% in SW480 cells and LS174T and ~95% in HCT116 tumor cells compared with their controls. A second dose, administered 72 h after the initial treatment, markedly increased cytotoxicity in normal cells, reducing viability to approximately 15%. This indicates that sequential dosing enhances cytotoxic effects in HCT116 and LS174T and was significant; it also compromises biosafety in normal cells (Figure 11).

## 3. Discussion

First of all, all cell lines were characterized in the study, confirming the absence of mycoplasma, as this can alter the results. All cell lines were capable of proliferating for approximately one week. SW480 and HCT116 cells exhibited similar morphology, characterized by adherent, epithelial-like shapes, whereas LS174T cells tended to form clusters or cell aggregates. In contrast, the normal colon cell line CCD18-Co displayed a smaller cell size and a distinct fibroblast-like morphology. These results are consistent with the American Type Culture Collection (ATCC) description that these cell lines came from.

On one hand, all cell lines were exposed to monotherapy with different concentrations of sodium butyrate [17,18,19] and bromelain [20] for 72 h, as this duration had been identified as the time point of the maximal response [19]. The dose–response curves indicated that CCD18-Co cells remained resistant to sodium butyrate (NaB) at concentrations up to 17.31 mM, whereas colorectal cancer cell lines exhibited a marked decrease in viability starting from as low as 1 mM in HCT116 cells, followed by SW480 (3.01 mM) and LS174T (7.33 mM).

Bromelain showed a dose-dependent cytotoxic effect in cancer cell lines, while normal colon cells (CCD18-Co) displayed greater resistance, especially at intermediate concentrations. This non-linear response may reflect an inverted U-shaped effect, where intermediate doses elicit maximal biological activity, whereas very high concentrations lead to enzyme autodegradation (autoproteolysis), diminishing bromelain’s functional efficacy. Previous work has shown that bromelain’s ability to remove specific cell-surface molecules depends on the structural properties of the target protein rather than the cell type itself [24]. This specificity may underlie the greater sensitivity of colorectal cancer cells, which often overexpress bromelain-sensitive adhesion and signaling molecules such as integrins and EGFR [25], compared with normal colon fibroblasts. Moreover, bromelain’s proteolytic integrity appears essential for effective cleavage, suggesting that autoproteolysis at high concentrations could limit its capacity to recognize and process relevant surface targets [24], contributing to the inverted U-shaped response observed in normal cells.

Furthermore, cancer cells are more reliant on hyperactivated survival pathways, such as EGFR signaling and integrin-mediated adhesion, which are susceptible to bromelain’s proteolytic action [24,25]. In addition, they typically exhibit higher baseline oxidative stress and a reduced capacity to manage proteotoxic damage [26]. In contrast, normal cells maintain more robust homeostatic mechanisms, including effective protein quality control, antioxidant defenses, and the ability to activate protective pathways such as autophagy [27]. These differences probably make cancer cells more vulnerable to stress caused by bromelain at intermediate concentrations, while normal cells can activate adaptive responses.

Numerous studies have demonstrated the anticancer properties of bromelain and sodium butyrate individually, showing effects on cell proliferation, apoptosis, epigenetic regulation, and tumor progression [6,7,8,9,10,11,12,13,14,18,19,20,21,22,25,26,28]. However, to the best of current knowledge, no published studies have investigated their combined use in cancer models.

Since tumors can develop resistance to single agents, drug combinations are an important therapeutic strategy for cancer. Targeting multiple biological mechanisms at once can improve treatment. Synergistic drug combinations may increase efficacy, allow lower doses, and reduce adverse effects [29,30,31]. For that, after stabilizing the dose–response curve, the IC_25_ value for the combination treatment was selected to analyze cell morphology, apoptosis–necrosis, cell cycle distribution, and gene expression in cancer viable cells. Initially, the IC_50_ was considered; however, the combined treatment with NaB and bromelain at their respective IC_50_ doses produced an additive effect, where 50% inhibition by NaB plus 50% inhibition by bromelain resulted in nearly 100% cell death. Since this additive effect leads to complete inhibition, it is unsuitable for detailed mechanistic analyses. Therefore, to study morphological changes, apoptosis–necrosis, cell cycle phases, and gene expression in treated tumor cells, as well as to evaluate synergy, a sublethal dose (IC_25_) [32,33] was used to preserve a viable cell population and capture these biological effects.

Colorectal cancer cell lines were analyzed following treatment with the IC_25_ combination. In the treatment with NaB, morphological alterations were observed in all tumor cell lines, and with bromelain the cells were detached as observed in other studies with cancer cells [34]. These observations were enhanced in the combined treatment, along with a reduction in the number of cells. There was an increase in both early and late apoptosis in cells treated with NaB [35], and this effect was enhanced in the combined treatment. The HCT116 cell line was the most affected, showing also a significant increase in necrosis, while the LS174T cell line was the least affected, likely due to its growth in cellular aggregates, which may limit drug penetration and reduce treatment efficacy. Ackerman et al. (2008) demonstrated that the formation of LS174T cell spheroids creates a physical barrier, resulting in low penetration of antibodies into the inner cells and potentially reducing the effectiveness of treatments [36].

A reduction in the number of tumor cells in the S phase was also observed in the treatment with NaB (SW480), which was further enhanced in the combined treatment, suggesting cell cycle arrest in the G1 phase [28,37]. This is consistent with the overexpression of *CDKN2A*, since the protein it encodes (p16INK4a) is known to induce cell cycle arrest by inhibiting CDK4 and CDK6, thereby halting progression from G1 to S phase [38].

On the other hand, when the IC_25_ combination (around 20 mM NaB and 20 µg/mL bromelain) was used in CCD18-Co cells, the cell viability decreased by about 25%, corresponding to 75% inhibition. Based on these observations, a dose–response matrix [39,40,41,42] was generated for all cell lines. The dose–response matrix showed that the normal cell line was more resistant to the combination than the colorectal cancer cell lines, and the optimal combination dose was 10 mM of NaB combined with 10 µg/mL of bromelain. At this concentration, viability in the normal cell line was approximately 75–80%, while it was only 5–15% in the cancer cell lines. Following the identification of the optimal dose, the cytotoxic effects of NaB, bromelain and its combination were evaluated in CCD18-Co. NaB and combination treatments reduced the proportion of cells in the S phase and induced early apoptosis, while overall cell viability remained around 75%. Notably, NaB induced cell cycle arrest in the G0 phase, suggesting a shift toward a quiescent, protective state in normal cells. *CDH1* (E-cadherin) expression was increased by NaB treatment and further enhanced in the combination group, indicating enhanced epithelial characteristics. This gene is essential for epithelial cell–cell adhesion and the maintenance of normal tissue structure suggesting a protective effect of NaB on normal colon cells [43]. Notably, synergistic effects in the cancer cells became evident at NaB concentrations of 10 mM, which also corresponded to the maximum tolerated dose that maintained acceptable viability in normal cells.

While the initial dose demonstrated selective cytotoxicity toward tumor cells with minimal effects on normal cells, the second dose administered after 72 h showed increased toxicity in normal cells (viability in the normal cell line decreased to 11%, while only 5% viability remained in the cancer cell lines), highlighting the need for careful optimization of dosing schedules to maximize therapeutic benefit while minimizing adverse effects.

The present study evaluated whether the combined treatment of NaB and bromelain could enhance anticancer effects in colorectal cancer cell lines. The combination induced apoptosis, G1 phase cell cycle arrest, and *CDKN1A* (p16) upregulation, while demonstrating limited cytotoxicity toward normal cells. These findings directly address the study objective, revealing that the combination provides enhanced anticancer effects compared with monotherapy.

Although previous studies have reported anticancer effects of NaB and bromelain individually [6,8,11,12,18,20,21,22], their combined effect in colorectal cancer models has not been explored. The observed synergy suggests that lower doses of each compound may achieve the desired therapeutic effects, potentially reducing toxicity and improving tolerability [29,44]. Moreover, investigating the recovery potential of residual cells could provide insights into resistance mechanisms and inform strategies to improve long-term treatment outcomes [45,46,47,48].

Despite the promising *in vitro* results, several limitations should be considered. First, all experiments were conducted in colorectal cancer cell lines, which do not fully capture tumor complexity *in vivo*, including interactions with the microenvironment or immune system [49,50,51]. Only a limited number of cell lines were examined, which may restrict generalizability to other colorectal cancer subtypes [52]. Furthermore, while low cytotoxicity was observed in normal cells, systemic toxicity and pharmacokinetic properties of the combination remain unknown. Addressing these limitations through 3D culture models, *in vivo* studies and detailed pharmacological analyses will be essential to determine translational potential [53,54].

Future research should aim to elucidate the molecular mechanisms underlying the synergistic interaction, optimize dosing schedules and delivery strategies, and validate the findings *in vivo*. Although previous *in vivo* studies have administered bromelain at doses of 3–10 mg/kg and NaB at approximately 600 mg/kg, the safety and efficacy of their combination have yet to be investigated [18,21,55].

If confirmed, the combination could be considered as an adjuvant to conventional chemotherapy, potentially enhancing efficacy, overcoming resistance, and improving patient outcomes. Additionally, the ability to target multiple cellular pathways may support the development of personalized therapeutic approaches based on specific tumor molecular profiles [56,57].

In summary, the findings suggest that combined treatment with sodium butyrate and bromelain enhances anticancer effects and may serve as a promising therapeutic approach to improve efficacy, reduce toxicity, and optimize patient response in colorectal cancer.

## 4. Materials and Methods

### 4.1. Cell Lines and Culture Conditions

The CC18-Co cell line isolated from healthy colon tissue and SW480, HCT116, and LS174T colorectal cancer cell lines were obtained from the American Type Culture Collection (ATCC, Manassas, VA, USA). Cells were maintained in Dulbecco’s Modified Eagle Medium (DMEM; Gibco, Paisley, UK) supplemented with 10% FBS (Gibco, Paisley, UK) and 1% Zellshield (Minerva Biolabs, Berlin, Germany). Cultures were incubated at 37 °C in a humidified atmosphere containing 5% CO_2_ until experiments. To characterize these cell lines, their morphology was analyzed using images obtained from cultures observed under a Zeiss Axio Vert A.1 optical microscope (Palex Medical, Madrid, Spain). Image acquisition and processing were performed using Zen 3.1 imaging software. To assess cell proliferation, Deep Blue reagent at 10% in culture medium (Invitrogen, Eugene, OR, USA) was applied to each cell line. The solution was added to cells before treatments and after 2 h of incubation at 37 °C and 5% CO_2_, 100 µL of solution was charged in a 96 well plate to measurement the changes in the fluorescence of Deep Blue solution.

### 4.2. Dose–Response Curve and IC_25_ Calculation

To calculate the IC_25_ (inhibitory concentration causing 25% reduction in viability) for each cell line, 4 × 10^5^ cells/cm^2^ were seeded at subconfluence. After 24 h to attachment, each cell line was treatment with from 1.25 to 20 mM [17,18,19] of Sodium Butyrate (NaB) (Sigma-Aldrich, St. Louis, MO, USA) and from 10 to 50 µg/mL [20] of Bromelain (Sigma-Aldrich, St. Louis, MO, USA) to calculate the doses witch the proliferation at IC_25_ after 72 h of treatment. To calculate IC_25_, GraphPad Prism Program 8 Version was used. In each cell line a control (cells culture in conventional medium, DMEM) was used.

### 4.3. Combination of IC_25_ Treatments

Once the IC_25_ values were calculated for each compound and cell line, a combination treatment using the IC_25_ concentrations of NaB and Bromelain was performed. Cells were seeded at a density of 4 × 10^5^ cells/cm^2^ at subconfluence. After 24 h to allow attachment, the cells were treated with the combined IC_25_ doses. Following 72 h of treatment, the cells were collected for further analysis.

### 4.4. Cell Viability Assay

This assay was performed following the manufacturer’s protocol for the Deep Blue reagent (BioLegend, San Diego, CA, USA) after treatments with NaB, bromelain, and their combination. A 10% solution of Deep Blue reagent was prepared in DMEM and added to the cells prior to treatments. After 2 h of incubation at 37 °C with 5% CO_2_, 100 µL of the solution was transferred to a 96-well plate to measure changes in fluorescence. Subsequently, 100 µL of medium from each sample were transferred to 96-well plates, and fluorescence was measured (excitation at 560 nm/emission at 590 nm) using a TECAN EnSpire multimode plate reader (Perkin Elmer, Waltham, MA, USA) with EnSpire Manager Software Version 4.

### 4.5. Flow Cytometry Analysis of Apoptosis, Necrosis, and Cell Cycle

After all treatments, the supernatant was collected, and adherent cells were detached using trypsin with 1X EDTA and combined into the same centrifuge tube. Cells were centrifuged at 200× *g* for 5 min and washed twice with 1X PBS before being transferred to 1.5 mL centrifuge tubes. The total cell suspension was divided into two tubes: one for apoptosis analysis and the other for cell cycle analysis.

For apoptosis detection, cells were washed twice with 1 mL of cold BioLegend’s Cell Staining Buffer (BioLegend, San Diego, CA, USA) and then resuspended in 500 μL of Annexin V Binding Buffer (BioLegend). Then, 100 μL of the cell suspension containing approximately 1 × 10^5^ cells was transferred to a new tube, and 1 μL of Annexin V-FITC (BioLegend) and 5 μL of 7-AAD (BioLegend) were added. The mixture was thoroughly mixed and incubated in the dark at room temperature for 15 min. Flow cytometry analysis was performed within one hour.

For the cell cycle assay, after washing twice with PBS, 3 mL of 70% cold ethanol was added dropwise to each tube for overnight fixation at 4 °C. The next day, cells were centrifuged at 200× *g* for 3 min to remove the ethanol, then washed twice with BioLegend Cell Staining Buffer. Subsequently, 1 × 10^5^ cells were resuspended in 100 μL of staining buffer, and 5 μL of 7-AAD (BioLegend) and 5 μL of Ki-67 (BioLegend) were added following the manufacturer’s instructions. Flow cytometry analysis was conducted within one hour.

Cells were acquired using a Cytek^®^ Northern Lights™ cytometer (Cytek Biosciences, Fremont, CA, USA). Data analysis was performed with FlowJo Software Version 10. Gating strategies involved selecting the main cell population based on forward and side scatter (FSC/SSC) and excluding doublets using singlet gating (FSC-A vs. FSC-H).

### 4.6. Analysis of Gene Expression

Total RNA was extracted using the NZY Total RNA Isolation Kit (NZYTech, Lisboa, Portugal) and quantified with a NanoDrop spectrophotometer (Thermo Fisher Scientific). One microgram of RNA was reverse-transcribed into cDNA using the High-Capacity cDNA Reverse Transcription Kit (Thermo Fisher Scientific, Baltics, UAB, Vilnius, Lithuania). Quantitative real-time PCR (qRT-PCR) was performed using PowerTrack™ SYBR Green Master Mix (Thermo Fisher Scientific, Baltics, UAB, Vilnius, Lithuania) according to the manufacturer’s instructions.

cDNA samples were run on a Thermal Cycler 7500 Fast Real-Time PCR System (Applied Biosystems, Foster City, CA, USA). Relative gene expression was calculated using the 2^−ΔCt^ method, with cycle threshold (Ct) values normalized to the expression of the glyceraldehyde 3-phosphate dehydrogenase (*GAPDH*) gene as the endogenous control. All samples were run in triplicate.

Primers for human *TP53*, *BAX*, *PTEN*, *CDH1* (E-cadherin), *CDKN1A* (Cyclin-Dependent Kinase Inhibitor 1A), and *CDKN2A* (Cyclin-Dependent Kinase Inhibitor 2A) were designed using the NIH Primer Design Tool https://www.ncbi.nlm.nih.gov/tools/primer-blast/ (accessed on 14 September 2025) following standard parameters. *GAPDH* (Glyceraldehyde-3-Phosphate Dehydrogenase) was used as the housekeeping gene for normalization (Table 1).

### 4.7. Synergy Analysis Between Sodium Butyrate and Bromelain

To analyze the synergy between NaB and bromelain, a dose–response matrix analysis was performed. NaB (1.25 to 20 mM) was combined with bromelain (10 to 50 µg/mL) across a range of concentrations for each cell line. Around 4 × 10^5^ cells/cm^2^ were seeded at subconfluence in 48-well plates, and after 24 h of attachment, cells were treated with the compound combinations. After 72 h of combined treatment, cell viability was assessed using 10% Deep Blue reagent in DMEM. Cells were incubated with the reagent for 2 h, and fluorescence was measured using a Tecan system. Viability was calculated and normalized to untreated control cells (set as 100% viability).

Synergy between treatments was evaluated using the SynergyFinder platform (SynergyFinder.org, version 3.0), applying the Highest Single Agent (HSA) model [58]. The HSA model was selected because it compares the observed combination effect to the effect of the most active single agent at the same concentration. This model is particularly appropriate when evaluating combinations of compounds that act through different biological pathways, as it does not assume any specific type of interaction between the agents. In this context, the HSA model allows us to determine whether the combined effect of NaB and bromelain is greater than their individual effects, even when they act independently.

### 4.8. Selective Cytotoxicity and Post-Treatment Recovery

For the selective lethal dose experiment, the minimal concentrations of each compound that preserved at least 75% viability in normal colon fibroblast cells (CCD18-Co), while inducing maximal inhibition in colorectal cancer cell lines, were selected based on dose–response matrix data. The selected combination was 10 mM of NaB with 10 µg/mL of bromelain.

Cells were seeded at subconfluence in 12-well plates and, after 24 h to allow attachment, treated with the selected combination. After 72 h of treatment, cell viability was assessed using the Deep Blue assay (BioLegend, San Diego, CA, USA). Viability was calculated and normalized to untreated control cells, which were set as 100% viable.

Normal cells (CCD18-Co) were also collected and analyzed to determine whether the selected concentrations exerted cytotoxic effects on non-tumorigenic cells, with the aim of confirming the safety profile of the combination therapy.

After the initial 72 h treatment, a second dose was administered to all cell lines. Subsequently, microscopic images were taken and cell viability was reassessed.

### 4.9. Statistic Analysis

Statistical analyses were performed using GraphPad Prism version 8. All experiments were carried out with three independent biological replicates, each with three technical triplicates. Results are presented as mean ± standard deviation (SD). Normality and homogeneity of variance were assessed using GraphPad Prism’s built-in tests. For statistical comparisons between two groups, Student’s *t*-test was applied. For comparisons among multiple groups, one-way ANOVA followed by Tukey’s post hoc test was used. Differences were considered statistically significant at *p* < 0.05, with α = 0.05. Statistical significance is indicated as follows: * *p* < 0.05; ** *p* < 0.01; *** *p* < 0.001; **** *p* < 0.0001.

## 5. Conclusions

The combined treatment of sodium butyrate and bromelain exhibited strong therapeutic synergy, beginning at 10 mM sodium butyrate and 10 µg/mL bromelain. This optimized single-dose regimen effectively targeted colorectal cancer cell lines while preserving approximately 75–80% viability in non-tumorigenic cells (CCD18-Co), demonstrating a favorable therapeutic window with single dose. These compelling results warrant further investigation with *in vivo* models to validate efficacy, pharmacokinetics, bioavailability, and long-term safety. Overall, this combination therapy represents a highly promising candidate for the development of innovative and effective pharmacological strategies against colorectal cancer, with significant potential to improve patient outcomes.

## Figures and Tables

**Figure 1 ijms-26-09803-f001:**
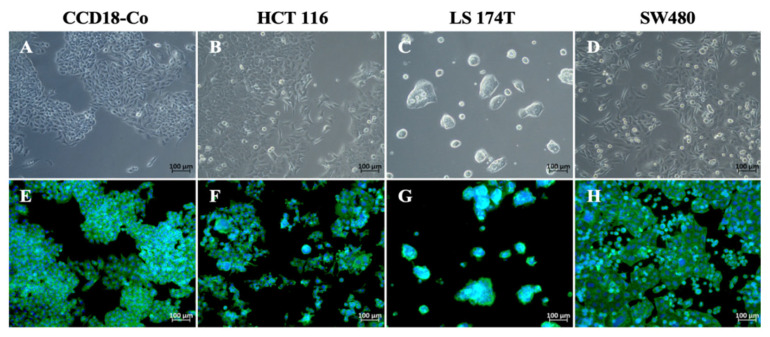
Morphological appearance in culture of normal human colon fibroblast cell line CCD18-CO (**A**,**E**), and colorectal cancer cell lines HCT116 (**B**,**F**), LS174T (**C**,**G**), and SW480 (**D**,**H**). Cells were observed under bright-field optical microscopy (**A**–**D**) and fluorescence microscopy after staining with DAPI (nuclei) and phalloidin (actin cytoskeleton) (**E**–**H**). Objective 20×.

**Figure 2 ijms-26-09803-f002:**
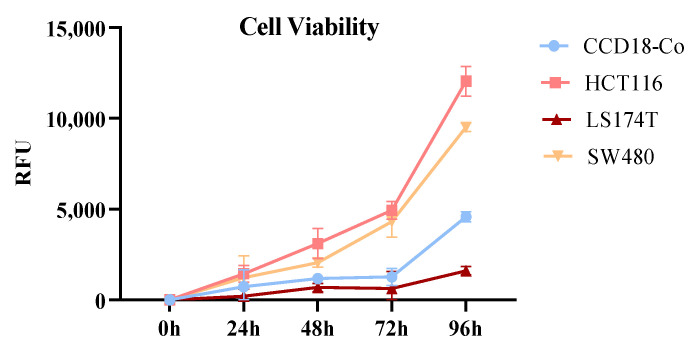
Cell proliferation of normal human colon fibroblast cell line (CCD18-Co) and colorectal cancer cell lines (HCT116, LS174T, and SW480) was analyzed. Experiments were performed in triplicate using the Deep Blue reagent (BioLegend) and Relative Fluorescence Units (RFU), following the manufacturer’s instructions. Data are presented as the mean of RFU ± SD.

**Figure 3 ijms-26-09803-f003:**
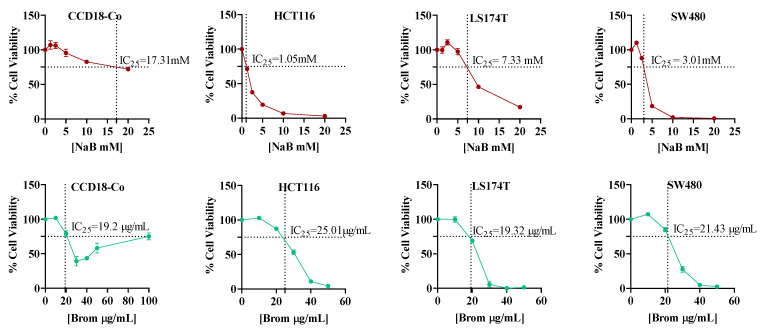
Dose–response curves showing the IC_25_ values (concentration causing 25% inhibition of cell viability) of Sodium Butyrate (NaB, red) and Bromelain (Brom, green) on CCD18-Co, HCT116, LS174T, and SW480 cell lines. Data were generated using GraphPad Prism version 8 and represent mean ± standard deviation from three independent experiments for each treatment and cell line.

**Figure 8 ijms-26-09803-f008:**
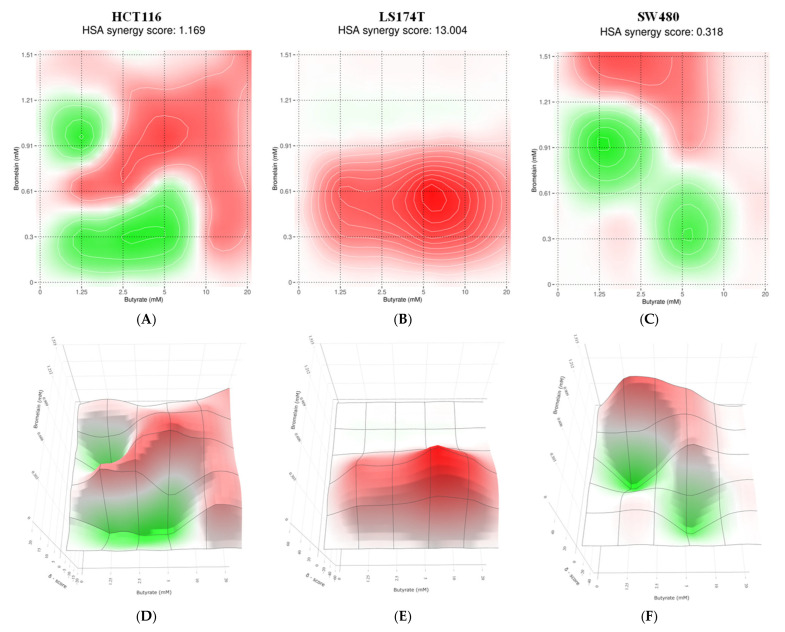
Two-dimensional (**A**–**C**) and three-dimensional (**D**–**F**) HSA synergy heatmaps generated with Synergyfinder.org show combinations of sodium butyrate (1.25–20 mM) and bromelain (10–50 µg/mL; concentrations expressed in mM) tested in three colorectal cancer cell lines: HCT116 (**A**,**D**), LS174T (**B**,**E**), and SW480 (**C**,**F**). Colors indicate the HSA δ-score, where red represents synergistic interaction and green indicates antagonism, with the scale ranging from −40 (strong antagonism) to +40 (strong synergy). Average HSA δ-scores for each cell line are shown above the corresponding 2D plots. Data represent the mean percentage of cell growth inhibition from a minimum of three independent experiments per condition.

**Figure 9 ijms-26-09803-f009:**
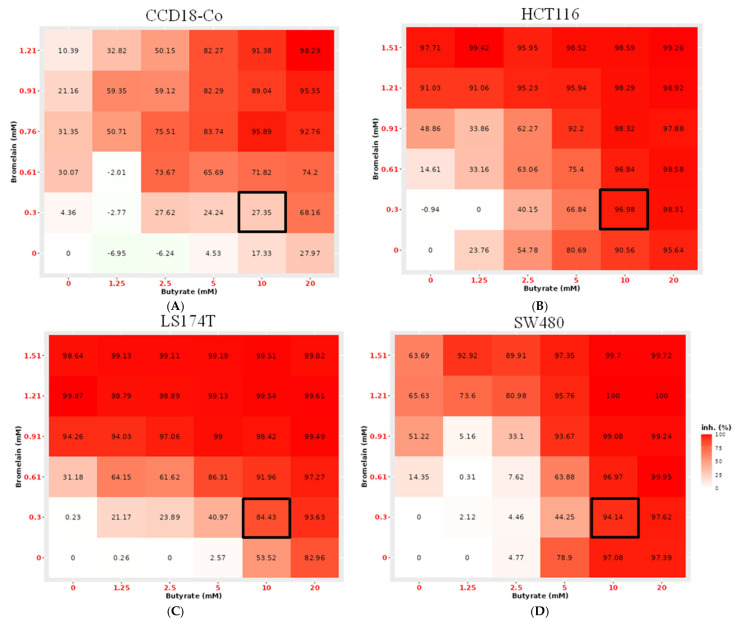
Dose–Response matrix showing the mean percentage of cell growth inhibition in the normal colon cell line CCD18-Co (**A**) and colorectal cancer cell lines HCT116 (**B**), LS174T (**C**) and SW480 (**D**) following treatment with combinations (expressed in mM) of sodium butyrate and bromelain. Analysis and visualization were performed using the SynergyFinder platform (synergyfinder.org). The black boxes indicate the optimal treatment concentration, which reduces tumor cell viability by over 80% while affecting healthy cells by only about 25%. Data represent the mean percentage of cell growth inhibition from a minimum of three independent experiments per condition.

**Figure 10 ijms-26-09803-f010:**
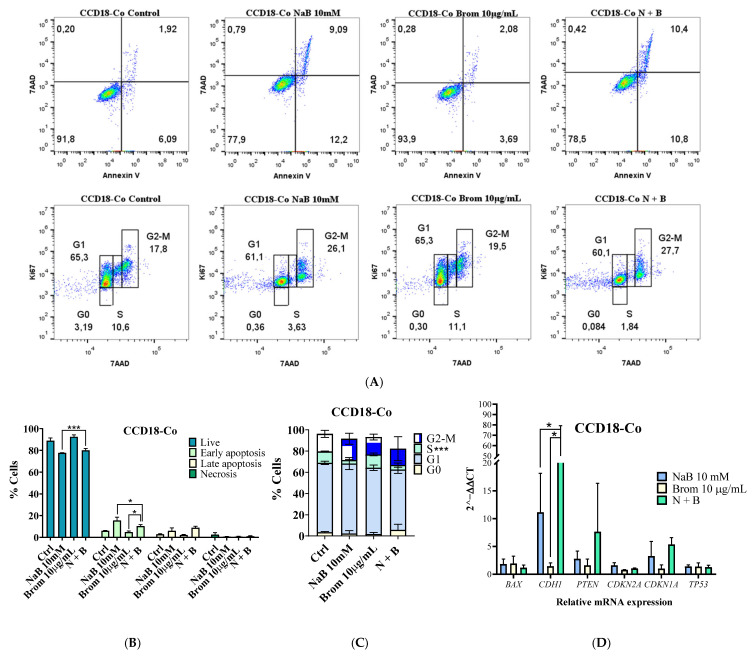
(**A**) Apoptosis, necrosis, and cell cycle flow cytometry data of CCD18-Co cells treated with 10 mM sodium butyrate (NaB), 10 µg/mL bromelain (Brom), or their combination (N + B). (**B**) Statistical analysis of apoptosis and necrosis and (**C**) cell cycle with the phases G0, G1, S, and G2–M of CCD18-Co cells after 72 h of treatment. The colors represent cell density, where blue corresponds to areas of low cell density and red indicates areas of high cell density. (**D**) Relative expression levels of *BAX*, *CDH1*, *PTEN*, *CDKN2A*, *CDKN1A*, and *TP53* in CCD18-Co cells after 72 h of treatment, using *GAPDH* as the housekeeping gene. The control group consisted of CCD18-Co cells cultured in conventional medium. Data represent the mean percentage of cell growth inhibition from a minimum of three independent experiments per condition. Statistical analysis was performed using one-way ANOVA followed by Tukey’s Post Hoc test. Significance: * *p* < 0.05; ** *p* < 0.01; *** *p* < 0.001; **** *p* < 0.0001; α = 0.05.

**Figure 11 ijms-26-09803-f011:**
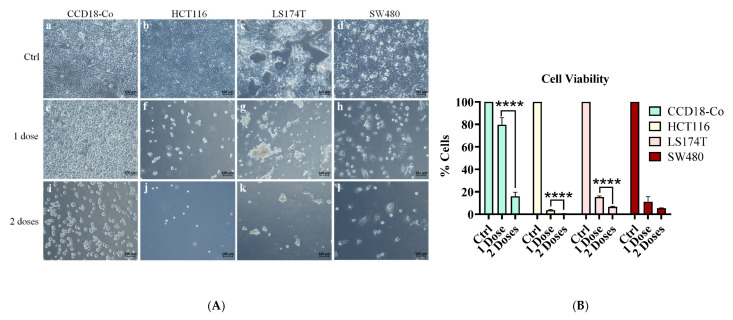
(**A**) Sequential treatment of a normal human colonic fibroblast cell line CCD18-Co (a,e,i) and colorectal cancer cell lines HCT116 (b,f,j), LS174T (c,g,k), and SW480 (d,h,l) cultured in conventional medium (Ctrl), after 72 h of combined treatment with 10 mM sodium butyrate and 10 µg/mL bromelain (1 dose), and after the first treatment followed by a second dose at 144 h (2 doses). (**B**) Cell viability assessed using Deep Blue after the first and second doses of the combination treatment (10 mM sodium butyrate + 10 µg/mL bromelain) in CCD18-Co, HCT116, LS174T and SW480 cell lines. Data are presented as mean ± SD from at least three independent experiments. Statistical analysis was performed using one-way ANOVA followed by Tukey’s Post Hoc test. Significance: * *p* < 0.05; ** *p* < 0.01; *** *p* < 0.001; **** *p* < 0.0001; α = 0.05.

**Table 1 ijms-26-09803-t001:** Sequences of Forward and Reverse Primers for Human Genes.

Gene	Gene Bank	Forward	Reverse
*BAX*	NM_138761.4	5′–CCACCAGCTCTGAGCAGATC–3′	5′–ACTCGCTCAGCTTCTTGGTG–3′
*CDH1*	NM_004360.5	5′–AAAGGCCCATTTCCTAAAAACCT–3′	5′–TGCGTTCTCTATCCAGAGGCT–3′
*CDKN1A*	NM_000389.5	5′–CCAACGCACCGAATAGTTACG–3′	5′–ACCAGCGTGTCCAGGAAG–3′
*CDKN2A*	NM_000077.5	5′–ATGGAGCCTTCGGCTGACT–3′	5′–GTAACTATTCGGTGCGTTGGG–3′
*GAPDH*	NM_002046.7	5′–GGTCACCAGGGCTGCTTTTA–3′	5′–TCGCCCCACTTGATTTTGGA–3′
*PTEN*	NM_000314.8	5′–AGACCAGTGGCACTGTTGTT–3′	5′–TCACCACACACAGGTAACGG–3′
*TP53*	NM_000546.6	5′–TCTGACTGTACCACCATCCACTA–3′	5′–CAAACACGCACCTCAAAGC–3′

## Data Availability

Data are contained within the article.

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
