# Peer review of "Combined Treatment of Sodium Butyrate and Bromelain Enhanced Anticancer Effects in Colorectal Cancer Cell Lines: A Promising Therapeutic Approach"

_ijms, 2025, doi:10.3390/ijms26199803_

Round 1

Reviewer 1 Report

Comments and Suggestions for Authors

First of all, I would like to congratulate the research team for producing this excellent research outcomes around cancer biology. This in vitro study explores a novel therapeutic strategy for colorectal cancer by combining sodium butyrate and bromelain. The results demonstrate encouraging potential for future development and clinical application. Overall, it is a well-conceived and promising piece of research that merits further investigation.

While the manuscript is well-structured and thoughtfully executed, there are few issues that can be addressed to make the manuscript better and are listed as below:

  1. Line 97: Missing period (.) at the end
  2. Line 103: It would be great if the authors could briefly share their rationale for including the cell proliferation data. If they feel it’s better suited for the supplementary data section, that’s totally fine too. Either way, a short explanation of its relevance would help readers understand how it helps with the flow of the overall findings.
  3. Line 110: Figure 2: Since it is said in legend the experiments were performed in triplicate, it would be helpful to include error bars to reflect data variability. If error bars were intentionally omitted, a brief rationale explaining this choice would improve transparency and help readers interpret the results more accurately.
  4. Line 121: It would be helpful if the authors indicated the numerical range of concentrations reported in literature, if feasible.
  5. Line 145: Instead of listing the sub-sections as bullet points, it may be more helpful to use numerical identifiers (e.g., 3.2.1). This would improve clarity and make it easier for readers to follow the structure and refer back to specific sections.
  6. Line 187: Figure 4: The figure legend appropriately mentions the necessary details, but it would be more informative and easier to follow if the cell line and treatment information were also included directly on or alongside the images, like what has been done in Figures 6A and 11, if feasible. This would help readers quickly interpret the data without needing to cross-reference the legend.
  7. Line 233: Table 1: The table appears somewhat redundant, as the statistical analysis indicators are already clearly presented in Figure 5. The authors might consider removing it to streamline the presentation, and instead simply mention the type of statistical analysis performed in the figure legend.
  8. Line 279: Figure 6 B, C and D: The way statistical analysis is shown using asterisks in the graph legends isn’t very clear. It might be easier for readers to follow if the significance indicators were presented in a more straightforward format, and their meaning explained more clearly in the figure legend.
  9. Line 289: Table 2 may be redundant, as long as the statistical analysis is clearly and easily indicated in Figure 6. If the figure conveys the necessary information in a straightforward way, the table might not add additional value.
  10. Line 302: Figure 7: It should be acceptable to indicate “mRNA expression” on the x-axis title and mention “2^ΔΔCT” in the figure legend, as this would clearly convey the quantification method used for the readers.
  11. Line 321: Figure 7: Mentioning the type of statistical analysis performed (e.g., ANOVA, t-test) in the legend of figures would be helpful for clarity and transparency.

Author response to REVIEWER 1

First of all, I would like to congratulate the research team for producing this excellent research outcomes around cancer biology. This in vitro study explores a novel therapeutic strategy for colorectal cancer by combining sodium butyrate and bromelain. The results demonstrate encouraging potential for future development and clinical application. Overall, it is a well-conceived and promising piece of research that merits further investigation.

While the manuscript is well-structured and thoughtfully executed, there are few issues that can be addressed to make the manuscript better and are listed as below:

RESPONSE REVIEWER 1:

We sincerely thank the reviewer for their valuable suggestions, which undoubtedly contribute to improving the quality of our manuscript. We are also pleased to know that our study was of interest. We will carefully address each of the comments point by point in our revised version.

  1. Line 97: Missing period (.) at the end

Response 1: We have added the period where it belongs. Thank you very much for the suggestion.

  1. Line 103: It would be great if the authors could briefly share their rationale for including the cell proliferation data. If they feel it’s better suited for the supplementary data section, that’s totally fine too. Either way, a short explanation of its relevance would help readers understand how it helps with the flow of the overall findings.

Response 2: We sincerely thank the reviewer for this constructive suggestion. As recommended, the following explanation has been included in the main text: “Cell proliferation data were included primarily to characterize the starting material. In addition, this analysis made it possible to compare the proliferative capacity of the different cell lines, to identify which line proliferates more rapidly, and to relate these differences to specific cellular characteristics¨.

  1. Line 110: Figure 2: Since it is said in legend the experiments were performed in triplicate, it would be helpful to include error bars to reflect data variability. If error bars were intentionally omitted, a brief rationale explaining this choice would improve transparency and help readers interpret the results more accurately.

Response 3: We thank the reviewer for this valuable observation. Error bars have now been included in Figure 2 to reflect data variability. We appreciate the reviewer’s attention to this detail, which indeed improves the transparency and clarity of the manuscript.

  1. Line 121: It would be helpful if the authors indicated the numerical range of concentrations reported in literature, if feasible.

Response 4: We have included the concentration ranges reported in the literature in the line 121, as you suggested:¨Cells were treated with a range of concentrations of NaB (1.25–20mM) and bromelain (10–50 µg/mL) based on values reported in the literature [17–20]. Although this information is already presented in the Materials and Methods section, we agree that adding it to the main text is helpful to the reader.

  1. Line 145: Instead of listing the sub-sections as bullet points, it may be more helpful to use numerical identifiers (e.g., 3.2.1). This would improve clarity and make it easier for readers to follow the structure and refer back to specific sections.

Response 5: We agree with your suggestion; however, the journal’s guidelines and the template require the use of bullet points for sub-sections.

  1. Line 187: Figure 4: The figure legend appropriately mentions the necessary details, but it would be more informative and easier to follow if the cell line and treatment information were also included directly on or alongside the images, like what has been done in Figures 6A and 11, if feasible. This would help readers quickly interpret the data without needing to cross-reference the legend.

Response 6: We have added the cell line and treatment information directly to Figure 4, as suggested. We believe this improves the clarity and overall quality of the manuscript. Thank you very much for your valuable suggestion.

  1. Line 233: Table 1: The table appears somewhat redundant, as the statistical analysis indicators are already clearly presented in Figure 5. The authors might consider removing it to streamline the presentation, and instead simply mention the type of statistical analysis performed in the figure legend.

Response 7: We have removed Table 1 as suggested, to streamline the presentation. The type of statistical analysis performed has now been clearly mentioned in the legend of Figure 5. We sincerely thank Reviewer 1 for this constructive suggestion.

  1. Line 279: Figure 6 B, C and D: The way statistical analysis is shown using asterisks in the graph legends isn’t very clear. It might be easier for readers to follow if the significance indicators were presented in a more straightforward format, and their meaning explained more clearly in the figure legend.

Response 8: We have added the following sentence to the legend of Figure 6 to improve clarity: “Significant differences are indicated with asterisks (*p < 0.05, **p < 0.01, ***p < 0.001, ***p < 0.0001; α = 0.05), representing comparisons between the combination treatment and the individual monotherapy treatments.”

This change makes it clear which comparisons are represented by the asterisks. We sincerely thank Reviewer 1 for this constructive suggestion, which has helped improve the clarity and readability of the figure and manuscript.

  1. Line 289: Table 2 may be redundant, as long as the statistical analysis is clearly and easily indicated in Figure 6. If the figure conveys the necessary information in a straightforward way, the table might not add additional value.

Response 9: We have considered Reviewer 1’s suggestion and agree that Table 2 is redundant. Therefore, we have removed Table 2, as the statistical analysis and significant differences are now clearly and directly indicated in Figure 6. We sincerely thank Reviewer 1 for this constructive suggestion, which helped us streamline the manuscript.

  1. Line 302: Figure 7: It should be acceptable to indicate “mRNA expression” on the x-axis title and mention “2^ΔΔCT” in the figure legend, as this would clearly convey the quantification method used for the readers.

Response 10: We have updated Figure 7 by adding "Relative mRNA expression" to the x-axes. Additionally, the figure legend has been revised to read: "Relative mRNA expression of BAX, CDH1, PTEN, CDKN2A, CDKN1A, and TP53 in colorectal cancer cell lines (HCT116, LS174T, and SW480) was quantified using RT-qPCR and analyzed using the 2^-ΔΔCt method." Thank you very much for your helpful suggestions.

  1. Line 321: Figure 7: Mentioning the type of statistical analysis performed (e.g., ANOVA, t-test) in the legend of figures would be helpful for clarity and transparency.

Response 11: We have updated the legend of Figure 7 and the rest of the manuscript by adding the following text: "Statistical analysis was performed using one-way ANOVA followed by Tukey’s post-hoc test."

We sincerely thank Reviewer 1 for their constructive suggestions, which have helped us make the manuscript more consistent, clearer, and easier to read.

Reviewer 2 Report

Comments and Suggestions for Authors

The article presents research on new therapies in cancer treatment, which is relevant as it focuses on gene expression and the efficacy of different treatment combinations. The methodology used is detailed, including the provision of materials and resources, as well as the study design and project supervision. The findings are illustrated through tables and figures showing significant results, although it is suggested that the clarity and citation of these elements in the text be improved. The author is advised to:

  • Include more context on the relevance of the study in the field of cancer research, as well as a more comprehensive review of previous studies supporting the need to investigate the combination of sodium butyrate and bromelain.
  • Results: it is important for the author to include a more detailed description of the most significant results, highlighting the most representative findings from the tables and figures, such as the gene expression results in different colorectal cancer cell lines, which can be seen in Figure 7. In addition, it may be mentioned that Table 3 presents the sequences of the primers used, which is crucial for the replication of the experiments. It is important to cite these figures and tables in the text, such as ‘as shown in Figure 7’ or ‘as detailed in Table 3,’ to guide the reader to the corresponding visual information.
  • Discussion: The author should address more explicitly the implications of the findings in the clinical context and how they compare with previous studies. It would also be useful to discuss the limitations of the study and suggest directions for future research.
  • To improve traceability, an explicit connection between the findings and the initial objective could be included. Remember that the title, objective, and conclusion should address the same criteria and variables considered in the study. Please ensure traceability in each section and conclude with respect to the stated objective.
  • The document complies with the journal's author guidelines.

Thank you

Author Response to REVIEWER 2

The article presents research on new therapies in cancer treatment, which is relevant as it focuses on gene expression and the efficacy of different treatment combinations. The methodology used is detailed, including the provision of materials and resources, as well as the study design and project supervision. The findings are illustrated through tables and figures showing significant results, although it is suggested that the clarity and citation of these elements in the text be improved. The author is advised to:

  1. Include more context on the relevance of the study in the field of cancer research, as well as a more comprehensive review of previous studies supporting the need to investigate the combination of sodium butyrate and bromelain.

RESPONSE REVIEWER 2:

We sincerely thank Reviewer 2 for taking the time to review our manuscript and for providing valuable suggestions. Your insights will undoubtedly help improve the quality of our work, and we will do our best to address your recommendations thoroughly

Response 1: To the best of our knowledge, no published studies have investigated the combined use of bromelain and sodium butyrate in cancer models. We have now included this point in the Discussion to acknowledge the current gap in the literature:

“Numerous studies have demonstrated the anticancer properties of bromelain and sodium butyrate individually, showing effects on cell proliferation, apoptosis, epigenetic regulation, and tumor progression [6–14,18–22,25,26,28]. However, to the best of current knowledge, no published studies have investigated their combined use in cancer models.”

We appreciate the reviewer’s suggestion, which helped improve the Discussion.

  1. Results: it is important for the author to include a more detailed description of the most significant results, highlighting the most representative findings from the tables and figures, such as the gene expression results in different colorectal cancer cell lines, which can be seen in Figure 7. In addition, it may be mentioned that Table 3 presents the sequences of the primers used, which is crucial for the replication of the experiments. It is important to cite these figures and tables in the text, such as ‘as shown in Figure 7’ or ‘as detailed in Table 3,’ to guide the reader to the corresponding visual information.

Response 2: We apologize for not being sufficiently clear in the original version and thank the reviewer for this helpful comment, which has improved the quality of the manuscript. The paragraph has been revised to provide a clearer description and better guide the reader:

¨After 72 h of IC25 combination treatment, colorectal cancer cell lines were harvested to analyze gene expression by RT-PCR. As shown in Figure 7, CDKN2A (p16) was upregulated in all cell lines treated with sodium butyrate (NaB), and this effect was further enhanced when combined with bromelain. The increase was statistically significant in SW480 cells, indicating a potential synergistic effect of the combination on cell cycle regulation. For further details, the primer sequences can be found in Table 3 of the Materials and Methods section.¨

  1. Discussion: The author should address more explicitly the implications of the findings in the clinical context and how they compare with previous studies. It would also be useful to discuss the limitations of the study and suggest directions for future research.

To improve traceability, an explicit connection between the findings and the initial objective could be included. Remember that the title, objective, and conclusion should address the same criteria and variables considered in the study. Please ensure traceability in each section and conclude with respect to the stated objective.

The document complies with the journal's author guidelines.

Thank you

Response 3:  We thank the reviewer for the valuable suggestions. We have expanded the Discussion to address all the aspects highlighted, including the comparison with previous studies, study limitations, potential mechanisms of synergy, and implications for future research and clinical applications.

The revised section now reads as follows:

The present study evaluated whether the combined treatment of NaB and bromelain could enhance anticancer effects in colorectal cancer cell lines. The combination induced apoptosis, G1 phase cell cycle arrest, and CDKN1A (p16) upregulation, while demonstrating limited cytotoxicity toward normal cells. These findings directly address the study objective, revealing that the combination provides enhanced anticancer effects compared with monotherapy.

Although previous studies have reported anticancer effects of NaB and bromelain individually [6,8,11,12,18,20–22], their combined effect in colorectal cancer models has not been explored. The observed synergy suggests that lower doses of each compound may achieve the desired therapeutic effects, potentially reducing toxicity and improving tolerability [29,44]. Additionally, investigating the recovery potential of residual cells could provide insights into resistance mechanisms and inform strategies to improve long-term treatment outcomes [45–48].

Despite the promising in vitro results, several limitations should be considered. First, all experiments were conducted in colorectal cancer cell lines, which do not fully capture tumor complexity in vivo, including interactions with the microenvironment or immune system [49–51]. Only a limited number of cell lines were examined, which may restrict generalizability to other colorectal cancer subtypes [52]. Furthermore, while low cytotoxicity was observed in normal cells, systemic toxicity and pharmacokinetic properties of the combination remain unknown. Addressing these limitations through 3D culture models, in vivo studies and detailed pharmacological analyses will be essential to determine translational potential [53,54].

Future research should aim to elucidate the molecular mechanisms underlying the synergistic interaction, optimize dosing schedules and delivery strategies, and validate the findings in vivo. Although previous in vivo studies have administered bromelain at doses of 3–10 mg/kg and NaB at approximately 600 mg/kg, the safety and efficacy of their combination have yet to be investigated [18,21,55].

If confirmed, the combination could be considered as an adjuvant to conventional chemotherapy, potentially enhancing efficacy, overcoming resistance, and improving patient outcomes. Moreover, the ability to target multiple cellular pathways may support the development of personalized therapeutic approaches based on specific tumor molecular profiles [56,57].

In summary, the findings suggest that combined treatment with sodium butyrate and bromelain enhances anticancer effects and may serve as a promising therapeutic approach to improve efficacy, reduce toxicity, and optimize patient response in colorectal cancer.

We believe that these additions improve the clarity and completeness of the Discussion and provide a more comprehensive perspective on the potential translational relevance of our findings.

We sincerely thank Reviewer 2 for highlighting these aspects, which have contributed significantly to improving the quality of our manuscript.